# Role of Thrombopoietin Receptor Agonists in Inherited Thrombocytopenia

**DOI:** 10.3390/ijms22094330

**Published:** 2021-04-21

**Authors:** José María Bastida, José Ramón Gonzalez-Porras, José Rivera, María Luisa Lozano

**Affiliations:** 1Department of Hematology, Instituto de Investigación Biomédica de Salamanca (IBSAL), Complejo Asistencial Universitario de Salamanca (CAUSA), Universidad de Salamanca (USAL), 37007 Salamanca, Spain; jrgp@usal.es; 2Department of Hematology and Oncology, Hospital Universitario Morales Meseguer, Centro Regional de Hemodonación, Universidad de Murcia, IMIB-Arrixaca, CIBERER-U765, 30008 Murcia, Spain; jose.rivera@carm.es (J.R.); mllozano@um.es (M.L.L.)

**Keywords:** inherited thrombocytopenia, thrombopoietin receptor agonists, MYH9-related disorder, Wiskott–Aldrich syndrome, platelet transfusions, bridging therapy

## Abstract

In the last decade, improvements in genetic testing have revolutionized the molecular diagnosis of inherited thrombocytopenias (ITs), increasing the spectrum of knowledge of these rare, complex and heterogeneous disorders. In contrast, the therapeutic management of ITs has not evolved in the same way. Platelet transfusions have been the gold standard treatment for a long time. Thrombopoietin receptor agonists (TPO-RA) were approved for immune thrombocytopenia (ITP) ten years ago and there is evidence for the use of TPO-RA not only in other forms of ITP, but also in ITs. We have reviewed in the literature the existing evidence on the role of TPO-RAs in ITs from 2010 to February 2021. A total of 24 articles have been included, 4 clinical trials, 3 case series and 17 case reports. A total of 126 patients with ITs have received TPO-RA. The main diagnoses were Wiskott–Aldrich syndrome, MYH9-related disorder and ANKRD26-related thrombocytopenia. Most patients were enrolled in clinical trials and were treated for short periods of time with TPO-RA as bridging therapies towards surgical interventions, or other specific approaches, such as hematopoietic stem cell transplantation. Here, we have carried out an updated and comprehensive review about the efficacy and safety of TPO-RA in ITs.

## 1. Introduction

Inherited thrombocytopenias (ITs) are a large heterogeneous group of rare platelet disorders characterized by low platelet counts (PC) which, not only lead to bleeding, but also to syndromic and systemic manifestations, and in some cases, they favor a predisposition to malignancy [1]. Their prevalence has been estimated to be around 1 in 100,000, but it is likely that this is an underestimate due to many individuals being undiagnosed [1,2]. The diagnosis is usually challenging and delayed. Patients can be misclassified as suffering from acquired disorders, especially when no other family members are affected, most frequently in de novo or recessive disorders [3]. In addition, the heterogenous and large number of genes being potentially involved increase the impediments to an ultimate definitive diagnosis. Since 2010, high throughput sequencing (HTS) methods have revolutionized DNA sequencing and the genetic diagnosis of human disease. The technique is now widely available in research and clinical practice and has been established as the gold standard for identifying the molecular pathology underlying monogenic diseases [2]. These techniques have allowed the identification of up to 40 genes as being causative of ITs [2,4], but despite the recent molecular progress in the diagnosis, in nearly 50% of patients with IT the genetic basis of these conditions remains unknown [5]. Less progress has been made in the therapeutic management of patients, which is based mainly on the transfusion of platelet concentrates [6]. The introduction of thrombopoietin (TPO) receptor agonists (RA) in the last ten years for different disorders, not only for immune thrombocytopenia (ITP), aplastic anemia (AA), but also for thrombocytopenia after hematopoietic stem cell transplantation (HSCT), open a new therapeutic possibility in ITs [7,8,9,10]. In this review we do not intend to update the advances in the molecular knowledge of ITs, which have been approached recently [4,5,11] but to raise the possibility of using this new class of agents, according to current evidence. We conducted a literature review of the MEDLINE, PubMed, and Cochrane databases using the subject headings ‘Inherited thrombocytopenia’, and the keywords ‘thrombocytopenia’, ‘thrombopoietin receptor agonists’, ‘thrombopoietin mimetics’, ‘romiplostim’, and ‘eltrombopag’ up to 12 February 2021. Each manuscript was screened by two reviewers (JMB and JRGP).

Our literature search identified a total of 24 studies with 126 patients diagnosed with IT and treated with TPO-RA as follows: 4 clinical trials, 3 series of cases and 17 case reports. Here, responses were defined as complete response (CR) when PC were >100 × 10^9^/L; partial response (PR), when PC were >30 × 10^9^/L and double of baseline platelet counts. Bleeding was recorded with World Health Organization (WHO) bleeding scale, or with the bleeding assessment tool of the International Society of Thrombosis and Hemostasis (BAT-ISTH) [12]. Adverse events (AEs) were recorded and graded according to the National Cancer Institute Common Terminology Criteria for AE, version 4.0 (https://evs.nci.nih.gov/ftp1/CTCAE/CTCAE_4.03_2010-06-14_QuickReference_5x7.pdf (accessed on 2 February 2021)).

## 2. Pathogenesis, Diagnosis and General Management of IT

### 2.1. Pathogenesis

The PC is the result of a balance between the biogenesis, senescence and consumption of these cells [11]. Megakaryopoiesis consists of different steps including the proliferation and differentiation of CD34+ hematopoietic stem cells (HSC) to megakaryocytes (Mks), Mk maturation and migration, proplatelet formation and the release of platelets to peripheral blood [11]. This whole process is highly regulated by transcription factors as well as by interactions with chemokines, cytokines and is also dependent on cytoskeletal proteins [11]. In this context, ITs are caused by genetic variants affecting genes which encode proteins involved in any steps of the process of megakaryopoiesis, causing thrombocytopenia with/without platelet dysfunction. Many ITs display large platelets while others have a normal or even small platelet size. While some cases exhibit isolated thrombocytopenias, many others are syndromic and/or associated with other conditions that may be of major clinical importance [11]. ITs may derive from defective commitment-differentiation of HSC to Mks leading to the absence or marked reduction of the number of Mks such as in congenital amegakaryocytic thrombocytopenia (CAMT) caused by mutations in *MPL* or *THPO* [5,11]. Defects in transcription factors that interfere with the early stages of Mk maturation are often accompanied by a risk of developing myeloid or lymphoid malignancies such as in *RUNX1*, *ANKRD26* or *ETV6* related disorders (RD) or related thrombocytopenia (RT) [5,11,13]. Decreased platelet counts not caused by defects in Mk differentiation and/or maturation, but rather by quantitative and/or qualitative proplatelet formation from mature Mks are the most frequent in our Spanish cohort (Figure 1). These ITs are related to cytoskeletal protein defects caused by mutations in genes encoding for components of the acto-myosin cytoskeleton (MYH9-RD, TUBB1-RT, ACTN1-RT, DIAPH1-RD or FLNA-RD, among others) or alteration of membrane glycoproteins (GP) that are predicted to affect cytoskeletal structure or reorganization, such as monoallelic Bernard–Soulier syndrome (mBSS) or ITGB3-RD [5,11,13]. There are other ITs such as Wiskott–Aldrich syndrome (WAS) where the decreased PC results from both an ineffective platelet production from premature proplatelet formation, and from increased platelet clearance. There are other mechanisms such as apoptosis (CYCS-RT), deficient glycosylation (GNE-RD and GALE-RD), mutations affecting ion gradient (STIM1-RD), increased tyrosine phosphorylation (SRC-RD), RNA degradation (SLFN14-RT) or poorly developed demarcation membrane system in Mks (sitosterolemia) [4,5,11,14,15,16]. The knowledge of these mechanisms could be the basis for the management of these complex disorders.

### 2.2. Diagnosis

In the majority of ITs, diagnosis is not straightforward or clinically unequivocal (such as in MYH9-RD, BSS or WAS). In the absence of a well-defined family history of thrombocytopenia, associated or not to bleeding, cases can still be confused mainly with ITP [3,11]. Diagnostic algorithms consist in a stepwise process based on clinical data and several laboratory tests. The widespread use of whole blood electronic counters to evaluate platelet count and size and blood smear with or without cytochemical or immunofluorescence staining are mandatory while assessing platelet function, is an important part of any strategy [17]. Classic tests to evaluate platelet function, such as aggregometry, secretion assays, flow cytometry, Western blotting and electron microscopy, are time consuming, expensive to perform and require trained personnel, and many of them are difficult to perform when the PC is low [11]. There is no doubt about the appropriateness of HTS procedures in the mainstream of diagnosis of platelet disorders including ITs [2,18,19]. However, the discussion is when and how to use these techniques due to the difficulties in the interpretation, the large amount of data that is generated, and the need for filtering and prioritization, or variants of uncertain significance [2,20,21]. Taking into account the variability in the availability of these tests, the fundamentals for a systematic approach to the diagnosis and management of ITs is necessary. For that, referral to a reference center may be desirable to decide the specific sets of tests to perform.

### 2.3. Management and Treatment

Unlike the progress in the diagnosis, especially at the molecular level, no important changes have been introduced in the clinical management of patients with ITs. Strategies consist of increasing PC, mitigating platelet dysfunction, and/or definitive correction of the ITs. Additional general measures contemplate platelet and red blood cell transfusions and the use of prohemostatic drugs, to prevent or treat bleedings. Allogeneic HSCT or gene therapy have the potential to be curative approaches in cases of severe ITs (Table 1 and Figure 2).

#### 2.3.1. General Measures

The prevention of bleeding is important for improving the general health of these patients [13]. A precise diagnosis allows in some cases a prediction of the severity of the disease, and patients can be informed about genetic counselling and family planification. Inappropriate treatments, such as corticosteroids, or splenectomy can be avoided. Drugs that impair platelet function, such as nonsteroidal anti-inflammatory agents, some antibiotics, cardiovascular, psychotropic, and oncologic treatments, agents that affect platelet cAMP, anesthetics, volume expanders, antihistamines, and radiographic contrasts can be prevented [2,20,22,23], or if needed, only administered after a careful assessment of the ratio between risks and benefits [6,24]. It is important to include patients on national and international registers, to increase the potential of reference centers for clinical research. One example is the RETPLAC (Registro Español de Trastornos Plaquetarios Congénitos, https://www.seth.es/index.php/investigacion/biologia-y-patologia-hemorragica/grupo-espanol-de-alteraciones-plaquetarias-congenitas-geap.html (accessed on 21 February 2021), a National registry that has been developed on behalf of the Spanish Society of Thrombosis and Hemostasis (SETH). Although the relationship between bleeding history with ISTH-BAT, laboratory or genetic testing and the future risk of bleeding does not always predict the bleeding risk [12,25], this information should be integrated in databases, and its analysis can constitute the basis for personalized counseling and management [26]. Moreover, detailed instructions should include measures for prevention or control of spontaneous bleedings such as epistaxis or gum bleeding and other at-risk situations, like dental and surgical procedures or traumatic events. In this context, regular dental care and oral hygiene are recommended to prevent gum bleeding and limit the need for invasive dental procedures [6,26]. Advice should be given where necessary about life-style issues (e.g., individuals with severe disorders should avoid contact sports) [6]. A special issue is bleeding in women (menarche, menstruation, pregnancy) which is usually a problem because untreated heavy menstrual bleeding frequently leads to iron deficiency and anemia, as well as to a decreased quality of life [27]. In these cases, multidisciplinary teams are recommended, and antifibrinolytics and/or hormonotherapy could be prescribed [27]. An important challenge is the preparation for delivery in which recently published guidelines recommend PC above 70 × 10^9^/L for neuraxial procedures (provided that platelet function is preserved) [28].

In cases of syndromic ITs, multidisciplinary teams are also helpful to define better management of other non-hemostatic manifestations of these disorders. Depending on the specific diagnosis, information on available treatments including antifibrinolytic agents, desmopressin, platelet transfusion, TPO-RA, recombinant activated FVII (rFVIIa), and replacement therapies should be provided (Table 1 and Figure 2).

#### 2.3.2. Prohemostatic Drugs

Desmopressin (1-deamino-8-D-arginine vasopressin, DDAVP) is used for managing bleeding episodes in patients with platelet function disorders and mild bleeding symptoms [29,30,31]. Since not all patients respond, a test dose is recommended to identify those who may benefit from receiving this agent for future spontaneous bleeding episodes. This agent is not recommended in patients with cardiovascular disease and infants below two years of age [13]. A multicentric retrospective worldwide study (SPATA study) has assessed the bleeding complications of surgery, the preventive and therapeutic approaches adopted and their efficacy in patients with inherited platelet disorders (IPDs) by rating the outcome of 829 surgical procedures carried out in 423 patients with well-defined forms of IPD (238 inherited platelet function disorders (IPFD) and 185 with ITs). The results showed that the use of preoperative prohemostatic treatments was associated with a lower bleeding frequency in patients with IPFD but not in ITs. DDAVP, alone or with antifibrinolytic agents, was the preventive treatment associated with lowest bleedings [32,33].

Antifibrinolytic (AF) agents, such as epsilon aminocaproic or tranexamic acid, are used by local application or systemic administration for prevention or treatment of mild to moderate bleeding episodes (especially useful for epistaxis, menorrhagia). In the SPATA study, AF agents were associated with a lower postsurgical bleeding frequency in ITs, while that was not the case for other treatments (except for desmopressin). AFs are usually contraindicated in the presence of hematuria [13,32].

rFVIIa, is a potent coagulation-enhancing treatment, that acts to amplify thrombin generation on the platelet surface, driving the formation of fibrin to improve platelet plug stability [29]. rFVIIa is approved for Glanzmann Thrombasthenia (GT) patients that have become alloimmunized and/or refractory to platelet transfusion. It has also been successfully used in BDPLT18 patients (GT-like or RASGRP2-RD) failing to respond to platelet transfusion [29,34,35]. Patients lacking the expression of specific GPs, such as GT or BSS are at higher risk of alloimmunization. It is preferable to avoid platelet transfusions in women of reproductive age with GT or BSS because if they develop antibodies to GPIb/IX or GPIIb/IIIa, these may transfer across the placenta during pregnancy and cause significant thrombocytopenia and bleeding in the fetus or in the new-born infant [13]. rFVIIa may also be considered as an off-label drug to be used in BSS since the absence of a major surface constituent (GPIb-IX-V) makes isoantibody formation likely, and in that circumstance, platelet transfusion ineffective [34,36].

#### 2.3.3. Platelet Transfusion

Platelet transfusion is a standard therapy to prevent or treat bleeding in thrombocytopenic patients, (Figure 2) [29]. The goal of platelet transfusion is to increase the number of healthy circulating platelets. One platelet unit (pooled platelet concentrates from whole blood donors or apheresis from a single donor) is expected to increase the peripheral PC by at least 30 × 10^9^/L. The circulating lifespan of transfused platelets may be similar to endogenous platelets in healthy subjects (6–8 days) [34]. Adverse effects include nonhemolytic febrile reactions, allergic reactions, transfusion purpura, infectious risks, acute transfusion-associated lung injury (TRALI), and alloimmunization. The development of antibodies against platelet specific antigens, which may be a big concern in patients with BSS (and GT), but also to human leucocyte antigen (HLA) antigens which, can lead to refractoriness to subsequent platelet transfusions [13]. To minimize the risk of alloantibody formation, platelet transfusions should preferably not only be leukocyte-depleted, but also HLA-matched. Since identifying potential donors for targeted donation of HLA matched platelets is sometimes not a simple and straightforward process, their use cannot be ensured in emergency situations, but rather in scheduled procedures [29]. To prevent alloimmunization, restrictive transfusion policies should be implemented to use these blood products only in severe bleedings that cannot be managed by local measures [6,13]. Once alloimmunization has been established, approaches such as continuous platelet infusion have been used to try to overcome this complication and treat active bleeding in hematological patients [37]. In this context, the need for prophylactic platelet transfusion before surgery should be carefully assessed in relation to the type of procedure and the patient’s features (PC and function, history of bleeding, overall evaluation of hemostasis) in order to avoid unnecessary transfusions. In fact, platelet transfusion, one of the most frequently used prophylactic treatments before surgery in ITs, has been revealed to be poorly effective, suggesting that either other treatments are required, or that the way platelet transfusions are employed (amount, type, timing) is inappropriate [32,33]. In this regard, dysfunctional platelets may impair the hemostatic function of healthy donor platelets, and the endogenous platelet count and the ratio of transfused versus endogenous platelets should be considered when treating selected IT patients with platelet transfusions [38].

#### 2.3.4. Splenectomy

Although, splenectomy has been effective in patients with WAS/X-linked thrombocytopenia (XLT), in terms of increasing PC and reducing the incidence of serious bleedings, as physicians have become more aware of the potential risks of subsequent severe infection, and recurrence of thrombocytopenia, the use of splenectomy is very rarely indicated in any form of IT [33,39].

#### 2.3.5. Curative Therapies

Curing the disease is still up for debate for ITs. HSCT is the treatment of choice for the most severe forms of ITs that endanger the lives of patients. It has been successfully used in children with WAS, CAMT and more recently in gray platelet syndrome (GPS) or GT [40,41,42]. However, a careful evaluation of the risk–benefit ratio must always be made [33]. Moreover, lentivirus-based gene therapy is already a proven therapy in WAS when there is not an optimal donor available for HSCT, although restoration of the platelet count may remain incomplete [43,44]. In the future, a risk profiling should be made case-by-case to identify candidates for gene therapy procedures, although currently this approach is not available for most forms of ITs, other than WAS [11,45].

## 3. TPO-RA and Rationale for IT Treatment

One highly promising, noninvasive approach to raising the platelet count is the use of TPO-RA [46]. TPO-RAs have been developed as a treatment option for several forms of thrombocytopenia [26]. There are various agents approved for ITP, such as romiplostim, eltrombopag and avatrombopag. Due to the longer time since the authorization of the first two, there is greater experience in their use as thrombopoietic agents in IT. These drugs bind to the TPO receptor causing its conformational change and leading to the activation of the JAK2/STAT5 pathway, which results in increased Mk progenitor proliferation and increased platelet production [7]. Romiplostim is a peptibody that binds directly and competitively to the TPO binding site, whereas eltrombopag and avatrombopag are small molecules which bind at a transmembrane site [7]. However, there are also differences in the activation of other signaling pathways such as STAT3, ERK and AKT [47]. Briefly, romiplostim mainly promotes proliferation of immature MKs through disproportionate activation of AKT with respect to the extracellular signal-regulated kinase (ERK)1/2 signaling pathway, whereas eltrombopag sustains proliferation and differentiation of Mks through balanced activation of both of these pathways [1,48,49]. With this rationale, TPO-RA could be effective in all forms of ITs where the main mechanism of thrombocytopenia is an impairment in the proplatelet formation and the in vitro differentiation, providing that maturation of Mks in response to TPO is preserved. In these circumstances, increasing the Mk mass could compensate for the reduced efficiency of platelet production [13,50]. In this context, one of the most studied IT is MYH9-RD, a situation where the differentiation and maturation of Mks in response to TPO is not impaired [13]. In physiological conditions, myosin IIA prevents premature release of proplatelets in the osteoblastic niche following Mk adhesion to type I collagen. In MYH9-RDs, these mechanisms are disrupted, causing an ectopic release of proplatelets and the generation of large platelets. The physiological process of platelet formation involves an important step of myosin IIA reactivation by shear stress, which determines the correct fragmentation and release of platelets in the vascular space. This shear reactivation is also lost in MYH9-RDs and causes a further release of large size platelets [51]. In this context, Romiplostim preferentially increases non-giant platelets, by interfering with this mechanism and inducing a predominant production of platelets of smaller size [52]. In DIAPH1-RD, there is an abnormal clustering of Mk and reduced proplatelet production, as well as abundant and disorganized actin, and the addition of eltrombopag improves these processes [53]. In WAS, the mechanism of thrombocytopenia is still open to debate, but evidence suggests that, similar to ITP, there is a combination of accelerated platelet destruction that leads to a shortened platelet lifespan as well as abnormal platelet production [39]. As TPO-RA have the ability to increase PC not only in ITP, but also in thrombocytopenia after HSCT, or aplastic anemia, this raises the hypothesis that these drugs can rescue the pathological phenotypes of ITs, mainly, caused by an impaired proplatelet fragmentation. Given that the differentiation and maturation of Mks is preserved in these forms of ITs, TPO-RA induce an increase in bone marrow Mks that overtake the reduced efficiency of platelet production and induce a predominant production of normal sized platelets [54]. In addition, it is important to highlight that some ITs derive from a variety of intrinsic genetic defects, which in some cases predispose to leukemic transformation, or the development of bone marrow fibrosis, premature cataracts, aspects to take into account for the possible use of TPO-RA in ITs [5,11,13].

## 4. Clinical Experience with TPO-RA for the Treatment of IT

The optimal management of patients with IT should start by providing specific care for their needs, but also to avoid overtreating those with little risk of bleeding. TPO-RA can be administered before elective surgery and/or invasive procedures, or prior to HSCT in cases of severe thrombocytopenia, such as WAS. In this situation, it is important to transiently increase the platelet count to >50 × 10^9^ platelets/L to reduce the risk of bleeding and the need for platelet transfusions.

### 4.1. Efficacy and Safety

It is difficult to compare the effectiveness of TPO-RA treatments in different forms of IT, considering that the various studies provide heterogeneous definitions of platelet responses. In Table 2, we refer to CR as PC > 100 × 10^9^/L; PR as PC > 30 × 10^9^/L and double of baseline counts. The studies also introduced in some cases the concept of bleeding responses, as the reduction or disappearance of bleeding symptoms, regardless of increases in PC.

To date, 126 patients with ITs have been treated with TPO-RA (Table 2, Table 3 and Table 4), with the majority of cases (116; 92%) having been included in 4 clinical trials (Table 2). Most enrolled patients were diagnosed as WAS/XLT (*n* = 81; 64%), MYH9-RD (*n* = 34; 27%), or ANKRD26-RT (*n* = 10; 8%) (Table 2).

In 2010, a first study evaluated the efficacy and safety of eltrombopag in 12 patients with MYH9-RD with PC < 50 × 10^9^/L. In this phase II, multicenter, open-label and dose-escalation trial, patients received 50 mg/day for 3 weeks, and those who did not achieve PC > 100 × 10^9^/L, were treated with an additional course of 75 mg/d for 3 weeks. If PC was more than 150 × 10^9^/L, eltrombopag was stopped. The primary endpoint was the ability to achieve PC > 100 × 10^9^/L or at least 3 times the baseline value (major response) or at least twice the baseline levels but less than major response (minor response). Secondary endpoints were safety, tolerability and reduction of bleeding tendency according to WHO bleeding score (grade 0: no bleeding; grade 1: petechiae; grade 2: mild blood loss; grade 3: gross blood loss; and grade 4: debilitating blood loss). According to the primary endpoint, 11/12 (92%) patients achieved any response: 8/12 (67%) achieved major response (five patients with 50 mg/day and three patients with 75 mg/day), while three patients attained minor response (25%); in only one patient the treatment resulted in no response. In those patients who achieved any response, PC still remained higher than baseline values 15 days after discontinuation of the drug. Other exploratory analyses showed no significant associations (PC, TPO levels, genetic variants in the motor or tail domains), except in the four splenectomized patients, who achieved major responses with PC higher than 100 × 10^9^/L. Bleeding disappeared in 8 of the 10 patients with hemorrhagic symptoms; the treatment was well tolerated and AE were mild and transient (headache and dry mouth) [55].

A second study (phase II) evaluated the use of eltrombopag on PC and activation in eight WAS/XLT patients. In this study, eltrombopag was dose-adjusted according to age with the aim to maintain PC > 50 × 10^9^/L (9–75 mg/d) during 22 to 209 weeks. The primary endpoint was achieving PC > 50 × 10^9^/L while the secondary outcomes were safety, platelet activation, and bleeding events according to WHO scale. Five out of eight patients (63%) achieved the primary endpoint, while 6/8 (75%) reduced the bleeding symptoms and only one patient showed mild and transient transaminitis. One of the nonresponder patients had a history of previous intracranial hemorrhage, one had been splenectomized and other had failed HSCT. One of these patients switched to romiplostim, achieving PC > 20 × 10^9^/L and decreasing the severity of bleeding [56]. Similar to ITP, eltrombopag did not increase platelet activation in WAS/XLT patients [57].

A phase II, multicenter, open-label and dose-escalation trial included the largest series of IT patients treated with eltrombopag. This trial consisted of two parts: the first aimed to analyze the short-term efficacy (6 weeks) of eltrombopag (50–75 mg/day) to achieve PC > 100 × 10^9^/L, and the second part was related to long-term (16 weeks) efficacy (25–75 mg/day) in reducing bleeding symptoms. In the first part, PC > 100 × 10^9^/L was considered a major response while a minor response was defined as the achievement of a platelet count at least two-fold higher than baseline without reaching major response. In the second part, a major response consisted of the complete resolution of bleeding, while a minor response was related to reduced bleeding symptoms. The secondary endpoints included safety and tolerability, improvement in quality of life, TPO levels and platelet function. A total of 24 IT patients were enrolled: MYH-9RD (*n* = 9); ANKRD26-RT (*n* = 9); WAS (*n* = 3); mBSS (*n* = 2) and ITGB3-RT (*n* = 1). The mean platelet count was 40 × 10^9^/L. One patient discontinued treatment early because of a grade 1 unlikely AE, in accordance with the protocol (increased plasma creatinine after 3 weeks of treatment). In the first part of the study, 21 out of 23 evaluable patients (91%) reached any response; 11 patients (48%) achieved major responses and 10 patients (43%) minor responses. Most MYH9-RD patients and mBSS had major responses while WAS and ANKRD26-RT showed minor responses. Only two patients (ANKRD26-RT and ITGB3-RT) did not respond. In the context of bleeding complications, 10/12 (83%) obtained complete resolution of symptoms. As expected, the two patients that failed to increase PCs (ANKRD26-RT and ITGB3-RT) did not improve the bleeding manifestations. Ten patients (43%) responded to 50 mg/day and stopped therapy while 13 patients (56%) required additional dose increase up to 75 mg/day. Five patients developed AE (*n* = 7) which were grade 1: mild and transient headache (*n* = 4) and/or diffuse bone pain (*n* = 3) during the first 3 days of treatment. In the second part of the study, four patients were enrolled to a long-term efficacy of eltrombopag (MYH9 = 2, WAS = 1, and ITGB3-RT = 1). Three patients completed the protocol and achieved stable complete response, while the WAS patient discontinued the treatment due to exacerbation of cutaneous eczema [46].

A recent retrospective, single-center, observational study from Russia (NCT04350164) analyzed the efficacy and safety of stable doses of romiplostim (9 mcg/kg/week) in 67 patients with WAS as a bridging therapy prior to HSCT. In this study patients were assigned a score system to define the severity (81% were considered classical WAS and 19% XLT). Bleeding was evaluated according to WHO bleeding scale. The efficacy was assessed and defined according to PCs as >100 × 10^9^/L (CR), increments > 30 × 10^9^/L from baseline to 100 × 10^9^/L (PR) and no response as not achieving increases in PC >30 × 10^9^/L above the baseline. The short-term efficacy was evaluated in the first 4 weeks and long-term response was assessed during months 2–12 of treatment. The study included 67 eligible WAS patients with ages ranging from 16 days to 14.9 years. The median pretreatment platelet count was 21 × 10^9^/L and the median treatment duration was 8 months (range 1–12 months). The efficacy was 60% for any type of response. Patients achieved CR after 1 week of therapy, and PR after 2 weeks. In responders, sustained responses were confirmed in the majority of cases (38/40). One patient with CR and one with PR lost platelet responses after 5 and 2 months of therapy, respectively. All patients had bleeding symptoms previous to TPO-RA, most of them were grade I/II gastrointestinal or nasal hemorrhages (58; 87%). Upon treatment, the bleeding tendency became negligible in responders (CR and PR), after one month of treatment. Interestingly, upon romiplostim treatment, patients in the non-responder group also demonstrated a decreased tendency toward clinically significant bleeding. According to AEs, only two patients developed thrombocytosis (>400 × 10^9^/L) and one patient with vasculitis presented arterial thrombosis, which was considered to be unrelated to treatment. No bone marrow abnormalities were found in the biopsies performed. The factors that predicted the absence of response were the severity of disease (WAS grade) and degree of thrombocytopenia (median PC = 16 × 10^9^/L; range 0–53) [39].

### 4.2. Management of ITs in the Surgical Setting

There are several case reports and one Italian series of cases that confirm the short-term efficacy and safety of TPO-RA for the perioperative increase of platelet counts in patients with ITs (Table 3).

The Italian prospective study analyzed the outcomes of 11 consecutive surgical procedures in 5 patients with MYH9-RD. Procedures included five major surgeries (orthopedic = 2, gynecologic = 1, and cochlear implantations = 2), four dental surgeries and a percutaneous kidney biopsy. Baseline PCs ranged from 5 to 25 × 10^9^/L and the ISTH-BAT bleeding score ranged from 3 to 22. It is important to note that three patients had previously undergone elective surgical procedures with prophylactic platelet transfusion, and that such procedures were complicated by significant bleeding episodes. Eltrombopag 75 mg/day was started 3 weeks before the procedure and continued 3 to 7 days after surgery. In 10/11 cases eltrombopag induced an increase in PC that allowed the execution of the surgical procedure without needing either platelet transfusions or having bleeding complications [58]. All of these cases achieved a major response in terms of PC > 100 × 10^9^/L or at least 3 times the baseline value [55]. In addition, the response continued for at least 5–7 days after eltrombopag discontinuation. Only one patient, with a deletion in exon 1 of MYH9 affecting the motor domain, did not respond and required platelet transfusion previous to the procedure. The patient presented a massive splenomegaly of unknown cause, which could justify the treatment failure.

Individual case reports are also detailed in Table 3. Surgical procedures included distal osteotomy, tympanoplasty, craniotomy and bilateral ovariectomy for MYH9-RD patients, lumbar recalibration for ANKRD26-RT and hip arthroplasty for DIAPH1-RD.

One patient received 50 mg/day of eltrombopag 20 days before a percutaneous distal osteotomy. Ten days after the start of treatment, the PC reached more than 100 × 10^9^/L, which facilitated the use of low molecular weight heparin (LMWH) following surgery. Before removal of the intramedullary Kirschner wire, eltrombopag was again used at a dose of 25 mg/day for 10 days, and the patient did again achieve a PC of 94 × 10^9^/L [59]. This agent has also been used in the pediatric population before a tympanoplasty procedure in a child with MYH9-RD, attaining pre- and peri-surgery PCs between 70 × 10^9^/L and 100 × 10^9^/L [60]. In MYH9-RD, surgical preparation using romiplostim is also feasible; a dose of 5 mcg/kg/week facilitated the increase in PC before a craniotomy for a cerebral aneurysm [52]. In the case of a disease that may predispose to the development of hematological neoplasms, such as ANKRD26-RT, the use of eltrombopag 50–75 mg/day allowed the performance of a lumbar recalibration in one patient [61]. The use of eltrombopag at 50 mg/d from 1 month before surgery, which was increased to 75 mg/day 1 week before the procedure, facilitated the performance of a right hip arthroplasty in a patient with DIAPH1-RD and a history of platelet alloimmunization and refractoriness [53]. Another type of complication to platelet concentrate transfusions was the development of previous allergic reactions in a patient with MYH9-RD. The prophylactic use of this blood product before a hysterectomy led to a serious reaction. Years later, eltrombopag 50 mg/day ensured a PC of 140 × 10^9^/L before surgery for a bilateral oophorectomy, and the patient could be treated with prophylactic LMWH after surgery. In this case, the MPV decreased from 20 fL to 15 fL and increased again 3 weeks after its discontinuation [54]. All the reported cases confirm the efficacy of the short-term use of TPO-RA to allow transient increases in platelet numbers in the perioperative setting, without major complications. We emphasize that with this approach patients neither required platelet transfusion nor rescue medications or procedures due to bleeding, and that to date no thrombotic complications have been described after the use of these agents in the surgical setting (Table 3) [58].

### 4.3. TPO-RA Use in IT Patients with Special Particularities

TPO-RA have been used in patients with an established diagnosis of IT under special conditions (pregnancy, chemotherapy, after failure of HSCT), and also in those with a previous misdiagnosis of ITP (Table 4). Eltrombopag was effective in a MYH9-RD patient to allow the administration of antineoplastic therapy for metastatic pancreatic cancer. The oral TPO-RA was started with the aim to achieve a platelet count higher than 80 × 10^9^/L, to allow therapy with 5-fluorouracil. At a dose of 75 mg/day, PC reached 185 × 10^9^/L. After 8 weeks of chemotherapy treatment, the patient was admitted for cholangitis and septic shock that required cholangiography with placement of a metallic stent. Although this complication was considered to be related to the cancer, eltrombopag was discontinued, and the PCs decreased to baseline values [62]. A case of special interest was that of a 41-year-old woman with MYH9-RD and a history of anaphylactic shock and angioedema following platelet transfusion and previous postpartum hemorrhage. Eltrombopag was started at 36 weeks’ gestation, achieving PC > 100 × 10^9^/L at 19 days. Both delivery and puerperium were uneventful, and LMWH was administered at prophylactic doses [63].

A previous misdiagnosis of ITP led to the treatment of a patient with MYH9-RD with romiplostim (fourth line therapy, after dexamethasone, intravenous immunoglobulin (IVIG) and mycophenolate mofetil) achieving a major response (PCs 3 times higher than the initial value) 2 months after the start of treatment. No AE were found during the 21 weeks of treatment, although a loss of response to the agent was evidenced at 5 months [64]. Similarly, a patient who was later classified as MYH9-RD received romiplostim after an incorrect diagnosis of ITP. He had previously shown a lack of response to corticosteroids and IVIG, and achieved a major response after 5 weeks of TPO-RA (3-fold increase from baseline). For 36 months, a mean platelet count of 33 × 10^9^/L was sustained. After one year of treatment, the dose of romiplostim was increased to 9 mcg/kg/week to prepare the patient for a surgical procedure, with PC reaching > 100 × 10^9^/L. During the 41 months of treatment of this patient with romiplostim, no AE or thromboembolic complications were recorded [65]. The third case of a patient with a misdiagnosis of ITP was that of a patient with severe macrothrombocytopenia and bleeding, who was treated with romiplostim (up to 10 mcg/kg/week) and developed hemorrhagic shock due to epistaxis. He was then treated with eltrombopag (50–75 mg/day) without an increase in PC, but with a reduction in mucocutaneous bleeding. After the failure of other lines of treatment for ITP, the patient was diagnosed with WAS and underwent HSCT reaching CR during a follow-up of more than 6 years [66]. Two remaining cases have been reported of patients classified as ITP before establishing a diagnosis of IT, who have been treated with TPO-RA. The first was a patient who achieved a CR for 13 months with eltrombopag 50 mg/day, before the diagnosis of DiGeorge syndrome. The other was a woman who received eltrombopag for 23 months before the diagnosis of Paris-Trousseau syndrome. Although this treatment was not associated with platelet responses, there was an evident reduction in bleeding complications, and no recorded AE during the time under therapy [67,68].

In two cases with a primary diagnosis of WAS, eltrombopag was administered in the peri-transplant setting. The first case was an 18-month-old boy who received the oral TPO-RA (0.8–5 mg/kg/day) for 32 weeks before HSCT. As in the previous case, although the PC did not increase, the agent was associated with a reduction in bleeding manifestations and in platelet transfusion requirements. In the second case, the child was treated with eltrombopag due to engraftment failure after HSCT; the TPO-RA was administered for 7 months, and no hematological responses were attained [67,69].

Romiplostim has been used in the treatment of five patients from three unrelated families with germline variants in the TPO gene (*THPO*), causing initial thrombocytopenia that evolved to inherited aplastic anemia. At doses of 4–5 mcg/kg/1–4 weeks, all patients experienced an increase in platelet count, disappearance of spontaneous bleeding, and transfusion independence. It is noteworthy that the lack of stimulation of the TPO receptor in this disease is associated with a reduction of the three hematopoietic lines, and that the use of romiplostim is associated with sustained trilinear hematological responses. One of the first reported patients had a brother who had required two HCSTs for bone marrow failure (BMF). Since the defect was not cellular but at the cytokine level (TPO), the procedures were unsuccessful and the patient died of severe gastrointestinal and pulmonary hemorrhage. The younger brother with the same clinical presentation was treated empirically with romiplostim, achieving a complete hematological response [70]. Similarly, a 3-month-old girl diagnosed with BMF in whom HSCT had failed, was treated with romiplostim, attaining also complete trilinear responses [70]. Three members of the same family were diagnosed with BMF due to a variant in *THPO*. They had recurrent major bleeding that required platelet transfusion. Romiplostim was initially administered at a dose of 1 mcg/kg/week for 3 months, with a rapid trilinear response and resolution of pancytopenia. Due to difficulties in the access to healthcare facilities, treatment was spaced to 4 mcg/kg each month, and with this regimen they have maintained a durable response during a 6-year follow-up [71].

## 5. Perspectives and Conclusions

In recent years, mounting evidence has been generated to support the use of TPO-RA to increase platelet counts in IT. IT encompasses a heterogeneous group of rare diseases in which platelet transfusion has long been considered the gold standard treatment. Data from four clinical trials, including more than 100 patients, confirm the efficacy and safety of TPO-RA as short- and long-term treatments in these disorders. Eltrombopag showed 92% platelet responses (67% CR) in MYH9-RD, 62% (CR 50%) in WAS, and 91% (CR 47%) in a set of variants of IT including MYH9-RD, ANKRD26-RT and others. In the case of WAS, 30% of patients had life-threatening bleeding, such as gastrointestinal and intracranial bleeding, as a consequence of thrombocytopenia [42,56].

Recently, romiplostim has shown 60% platelet responses (CR 33%) as bridging therapy in patients with WAS before HSCT. Additionally, in patients who did not increase PC, the hemorrhagic phenotype was markedly reduced after TPO-RA therapy. Furthermore, no major AEs such as bone marrow fibrosis, malignancies, or thrombosis were described. There does not seem to be a correlation between the genetic defect and the probability of response.

These agents can also be used to prepare patients for elective invasive procedures or surgeries. To date, they have been used in 19 procedures, being effective in all cases except one. Most of the patients suffered from MYH9-RD (*n* = 17, 89%), and in most cases eltrombopag was started 1 month before the procedure (*n* = 17, 89%). Although there is heterogeneity in the doses used, patients generally responded within 3 weeks. Based on existing data and expert recommendations, patients with IT undergoing surgery should be offered an off-label treatment for 3 weeks with 50 mg/day of eltrombopag with the option of another three weeks at 75 mg/day in those who do not respond adequately [45,46,58]. Despite the limited data, there is no evidence to think that romiplostim would be less effective or more toxic than the oral TPO-RA. The addition of LMWH after surgical procedures in patients receiving TPO-RA has only been used in two cases. This is consistent with the SPATA study, in which antithrombotic prophylaxis is rarely used (10.5%) in patients with congenital platelet abnormalities after invasive procedures [72].

In patients with WAS who are scheduled to undergo HSCT, romiplostim has been shown to be effective as a bridging therapy to allow the administration of conditioning therapy. In previous studies, the overall platelet response rate is around 60%, with a significant reduction in bleeding symptoms and in the need for platelet transfusion, even in patients who do not achieve platelet responses [39]. Furthermore, romiplostim is effective and safe in thrombocytopenic patients who develop BMF due to *THPO* mutations [70,71].

The risk of clonal evolution and malignancy and that of bone marrow fibrosis is a clinical concern with the use of TPO-RA. However, the evidence that TPO-RA accelerates disease progression in patients with myelodysplastic syndrome or acute leukemia is controversial [9,45,73]. To date, these complications have not been observed in patients with IT, although only 10 patients with ANKRD26-RT have been treated with TPO-RA. There is also no evidence of the development of marrow abnormalities in 22 patients with WAS treated with romiplostim [39], nor have thrombotic complications been described to date. Although TPO-RAs have not been approved for IT, the data support that their use, in most cases as short-term therapies, is beneficial in terms of efficacy and safety, and may result in the improvement of the patient’s quality of life. The risk-benefit balance must be individually assessed in the face of certain associated disorders such as cancer, atrial fibrillation and other age-related comorbidities.

In conclusion, the existing data indicate that TPO-RA can be an effective treatment, especially in those IT-related defects in cytoskeletal assembly, proplatelet formation or in cases of reduced platelet lifespan. In spite of a well-established predisposition to hematological neoplasms in patients with ANKRD26-RT, short courses of treatments with TPO-RA are considered to be safe in this disease.

## Figures and Tables

**Figure 1 ijms-22-04330-f001:**
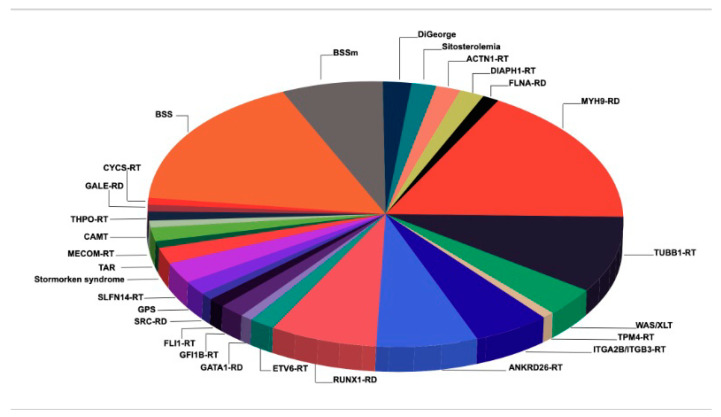
Percentages of patients with inherited thrombocytopenias, with a confirmed molecular diagnosis, identified among 114 families in Spain. Abbreviation: RT: related thrombocytopenia: RD: related disorder; BSS: Bernard–Soulier syndrome; m: monoallelic; TAR: thrombocytopenia absent radii; CAMT: congenital amegakaryocytic thrombocytopenia; GPS: gray platelet syndrome; WAS/XLT: Wiskott–Aldrich syndrome/X-linked thrombocytopenia.

**Figure 2 ijms-22-04330-f002:**
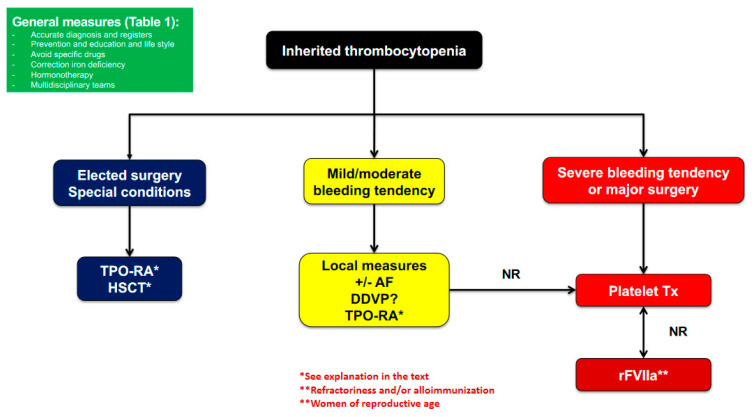
General algorithm to the management of thrombocytopenia and bleeding in IT. TPO-RA could be a reasonable option for IT with cytoskeleton protein alteration and proplatelet formation, or reduced life span. Short-term eltrombopag is a valuable treatment for MYH9-RD in reduced bleeding symptoms and before surgery. Romiplostim could be used for a bridging therapy in WAS patients before HSCT. HSCT is considered a curative option in severe forms of IT. Abbreviations: AF: antifibrinolytics; TPO-RA: Thrombopoietin receptor agonist; Tx: transfusions; HSCT: hematopoietic stem cell transplant; DDVP: desmopressin; rFVIIa: recombinant activated factor VII.

**Table 1 ijms-22-04330-t001:** General measures and treatment options in the management of patients with inherited platelet disorders, with a special focus in those with inherited thrombocytopenias.

Educational measures	Avoid trauma or bleeding risk situations (sports with strong contact or high risk of falls). Appropriate oral hygiene. Personal identification of the disorder (identification plates, etc.). Avoid drugs and foods with antiplatelet potential.	
Preventive measures	Inclusion in registries. Hepatitis B vaccination. Assessment of liver function at diagnosis. Routine medical follow-up, including asymptomatic patients, but with disorders at high risk of developing syndromic pathology or neoplasms. Specialized and multidisciplinary care for hemorrhagic risk interventions. Prenatal diagnosis and bleeding prevention plan in childbirth, surgeries, dental interventions, or antiplatelet treatments due to cardiovascular risk.	
Control of moderate active bleeding Prevention of bleeding in low hemorrhagic risk interventions Prophylactic treatment in patients with moderate disorders	Topical Measures	Mild wounds: compression and application of gelatin sponges or gauzes soaked in tranexamic acid Epistaxis/gum bleeding: nose pads, fibrin sealants, fibrin-coated/collagen sponges, mouthwash with tranexamic acid, anterior or posterior packing. Dental interventions: splint of soft acrylic assist, fibrin sealants.
	Antifibrinolytic drugs	Control of mild—moderate bleeding (epistaxis or menorrhagia) and in prevention of bleeding in minor dental interventions. Treatment begins the day before and continues for 3–5 days. Dose: Tranexamic acid: oral 15–25 mg/kg/8 h; iv 10 mg/kg/8 h if more severe bleeding; for mouthwash (10 mL at 5%); e-aminocaproic acid: oral 4–8 g/8 h; iv 4 g/8 h
	Desmopressin (DDAVP)	Few data in inherited thrombocytopenia and not routinely recommended. Not recommended in PT-VWD, or in patients with atherosclerosis. Drug of choice in patients with mild/moderate IPFD undergoing dental interventions and minor surgeries. Causes fluid retention. Risk of desensitization with repeated doses. Dose: I.V. (0.2–0.3 ug/Kg of DDVAP in saline [4 ug/mL] in 30 min, start 1 h before); subcutaneous (0.3 ug/kg); Intranasal spray (150 ug/dose)
Control of moderate to severe bleeding, prevention of bleeding in high bleeding risk interventions, preventive treatment in patients with severe disorders	Platelet transfusion	Essential for control of severe bleeding, in severe thrombocytopenia, for prevention of bleeding in major surgery and for the management of childbirth in a woman with severe platelet dysfunction. Preferable leucocyte-depleted platelet concentrates from single donor and/or HLA identical if possible, to reduce the risk of alloimmunization or if the patient already has anti-HLA antibodies; Recommended platelet counts: (If platelet dysfunction or severe bleeding history, individualization is mandatory). >30 × 10^9^/L for dental extractions and minor dental interventions; >50−80 × 10^9^/L for major surgery, deliveries, or caesarean sections; >100 × 10^9^/L for eye and brain surgery.
	Recombinant active Factor VII (rFVIIa)	Approval in GT patients with platelet transfusion refractoriness. Off-label use in other IPDs. May also be considered as an off-label drug to be used in BSS (risk of alloantibodies and ineffective platelet transfusion. Potentially useful, in combination with antifibrinolytics, in the control of bleeding in childbirth of women with severe platelet dysfunction. Dose: I.V. 90–120 ug/Kg before the procedure and then repeated doses every 90–120 min. The required number of doses is variable depending on the risk of the procedure and the patient characteristics.
Increase of platelet count stably or transiently before surgery or invasive interventions	Splenectomy	To be considered only in WAS and XLT and a patient personalized basis. May reduce bleeding complications but worsens the immunodeficiency and the rate of severe infections. It should be avoided in WAS patients who have undergone or are candidates of HSCT.
	TPO-RA (Eltrombopag, Romiplostim)	To be considered for short-term use (for instance to increase in numbers before elective surgery) in WAS, MYH9-RD, or ANKRD26-RT. Long-term treatment has been successfully used in some patients with MYH9-RD and WAS Romiplostim was successfully used in THPO-RT.
Potential curative treatments	Allogeneic hematopoietic stem cell transplantation	To be considered in severe ITs at high risk of transformation to bone marrow failure or malignant disease and with high early mortality. Treatment of choice in CAMT and severe WAS. Successfully used in some severe cases of TAR, RUSAT, BSS and GT (about 60 cases).
	Gene therapy	Clinical trial in WAS. Preclinical studies in other IPD (mainly GT).

Abbreviations: VWF: von Willebrand factor; PT-VWD: platelet-type von Willebrand disease; HLA: human leucocyte antigens; IPFD: inherited platelet function disorders; GT: Glanzmann thrombasthenia; IT: inherited thrombocytopenia; IPD: inherited platelet disorders; TPO-RA: Thrombopoietin receptor agonists; WAS: Wiskott–Aldrich syndrome; XLT: X-linked thrombocytopenia; MYH9-RD: MYH9-related disease; ANKRD26-RT: Ankyrin repeat domain 26 related thrombocytopenia; CAMT: congenital amegakaryocytic thrombocytopenia; TAR: thrombocytopenia with absent radii; RUSAT: radio-ulnar synostosis with amegakaryocytic thrombocytopenia; BSS: Bernard–Soulier syndrome. HSCT: hematopoietic stem cell transplantation; most of these measures are also useful for IPFDs.

**Table 2 ijms-22-04330-t002:** Clinical trials including patients with inherited thrombocytopenia treated with TPO-RA.

Disease	Type of Study	TPO-RA	Dose	N	Mean PC	Indication	Type of Response	Treatment Duration	Adverse Events (n)	Ref
MYH9-RD	Phase II	Eltrombopag	50–75 mg/d	12	31.2 × 10^9^/L	Efficacy and safety of short-term course	R: 92% (CR: 67%) B: 80%	3–6 w	Headache (2) Dry mouth (1)	[55]
WAS/XLT	Phase II	Eltrombopag	0.8/kg/d–75 mg/d	8	19 × 10^9^/L	Efficacy and safety	R: 62.5% (CR: 50%) B: 75%	20–187 w	Transaminitis (1)	[56]
MYH9-RD (9) ANKRD26-RT (9) WAS/XLT (3) mBSS (2) ITGB3-RT (1)	Phase II	Eltrombopag	25–75 mg/d	24	40 × 10^9^/L	Efficacy and safety of short and long-term course	R: 91.3% (CR: 47%) B: 83%	3–6 w	Headache (4) Bone pain (2) Creatinine, nr (1)	[46]
WAS/XLT	Observational	Romiplostim	9 mcg/kg/w	67	21 × 10^9^/L	Efficacy and safety for bridging to HSCT	R: 60% (CR: 33%) B: 100%	1–12 m	Thrombocytosis (2) Arterial thrombosis, nr (1)	[39]

Abbreviations: N: number of patients included; y: years; F: female; M: male; PC: platelet counts; bleeding was defined according to WHO bleeding scale; B: bleeding response; w: weeks; d: day; m: months; Ref: reference; RD: related disorder; WAS: Wiskott–Aldrich Syndrome; XLT: X-linked thrombocytopenia; RT: related thrombocytopenia; mBSS: monoallelic Bernard–Soulier Syndrome; nr: not related; HSCT: hematopoietic stem cell transplantation; CR: complete response; R: platelet response (partial and complete platelet responses).

**Table 3 ijms-22-04330-t003:** Previous experience with the use of TPO-RA to prepare patients with inherited thrombocytopenia for elective surgeries and/or invasive procedures.

Disease (n)	Type of Study	TPO-RA	Dose and Weeks before Surgery	Mean PC	Surgery	PC Day of Surgery (d)	Time to Response	Adverse Events	Ref
MYH9-RD (1)	Case report	Eltrombopag	50 mg/d (3 w)	19 × 10^9^/L	Osteotomy	195 × 10^9^/L (19) CR	10–12 d	No LMWH	[58]
MYH9-RD (1)	Case report	Eltrombopag	25 m/d (1 w) 50 mg/d (3 w)	10 × 10^9^/L	Tympanoplasty	77 × 10^9^/L (33) PR	33 d	No -	[59]
MYH9-RD (1)	Case report	Romiplostim	1–5 mcg/kg/w (5 w)	25 × 10^9^/L	Craniotomy	84 × 10^9^/L (42) PR	6 w	No -	[52]
ANKRD26-RT (1)	Case report	Eltrombopag	50 mg/d (4 w) 75 mg/d (1 w)	16 × 10^9^/L	Lumbar recalibration	93 × 10^9^/L (35) PR	2 w (lost) 5 w	No LMWH not reported	[60]
DIAPH1-RD (1)	Case report	Eltrombopag	50 mg/d (3 w) 75 mg/d (1 w)	29 × 10^9^/L	Hip arthroplasty	72 × 10^9^/L (28) PR	3 w	No LMWH not reported	[53]
MYH9-RD (1)	Case report	Eltrombopag	50 mg/d (4 w)	32 × 10^9^/L	Ovariectomy	153 × 10^9^/L (28)	2 w	No LMWH	[54]
MYH9-RD (1)	Case report	Eltrombopag	75 mg/d (-)	20 × 10^9^/L	Urgent endoscopic treatment	93 × 10^9^/L (-) PR	-	No	[61]
MYH9-RD (1)	Case report	Romiplostim	9 mcg/kg/w (5 w) + Prednisone	6 × 10^9^/L	General	115 × 10^9^/L (42) CR	5 w	No	[64]
MYH9-RD (5)	Case series	Eltrombopag	50 mg/d (3 w)	19 × 10^9^/L	Osteotomy	180 × 10^9^/L (20) CR	3 w	No LMWH not reported	[57]
				20 × 10^9^/L	Osteotomy	172 × 10^9^/L (21) CR	3 w	No LMWH not reported	
				23 × 10^9^/L	Percutaneous kidney biopsy	161 × 10^9^/L (21) CR	3 w	No LMWH not reported	
			75 mg/d (3 w)	15 × 10^9^/L	Hysterectomy and bilateral annexectomy	75 × 10^9^/L (21) PR	3 w	No LMWH not reported	
				17 × 10^9^/L	Cochlear implantation	78 × 10^9^/L (22) PR	3 w	No -	
			75 mg/d (3 w)	710^9^/L	Dental extraction	100 × 10^9^/L (21) CR	3 w	Headache -	
				9 × 10^9^/L	Periodontal surgery	120 × 10^9^/L (21) CR	3 w	Headache -	
				10 × 10^9^/L	Dental extraction	95 × 10^9^/L (21) CR	3 w	No -	
				10 × 10^9^/L	Periodontal surgery	132 × 10^9^/L (22) CR	3 w	No -	
			75 mg/d (3 w)	25 × 10^9^/L	Cochlear implantation	104 × 10^9^/L (23) CR	3 w	No -	
			75 mg/d (3 w)	5 × 10^9^/L	Biopsy of tonsillar tumor	11 × 10^9^/L (21) NR	No	No -	

Abbreviations. RD: related disorder; n: number of patients included; y: years; F: female; M: male; PC: platelet counts; bleeding was defined according to WHO bleeding scale; w: weeks; d: day; Ref: reference; LMWH: low molecular weight heparin; CR: complete response (defined by PC > 100 × 10^9^/L); PR: partial response (defined by PC > 30 × 10^9^/L and/or double of baseline platelet counts); NR: no response.

**Table 4 ijms-22-04330-t004:** Case reports and series of cases of patients with inherited thrombocytopenia treated with TPO-RA were used in special situations.

Disease (n)	Type of Study	TPO-RA	Dose	Mean PC	Special Situation	Type of Response	Treatment Duration	Adverse Events (n)	Ref
MYH9-RD (1)	Case report	Eltrombopag	50–75 mg/d	20 × 10^9^/L	Chemotherapy Endoscopic treatment	CR	2 m	No	[61]
MYH9-RD (1)	Case report	Eltrombopag	50 mg/d	30 × 10^9^/L	Gestation	CR	24 d	No LMWH	[62]
MYH9-RD (1)	Case report	Romiplostim	10mcg/kg/w	7 × 10^9^/L	Misdiagnosis of ITP (2nd lines)	60 × 10^9^/L PR	21 w	No	[63]
WAS (1)	Case report	Romiplostim and Eltrombopag	10 mcg/kg/w 25–75 mg/d	5 × 10^9^/L	Misdiagnosis of ITP (2nd lines)	NR Bleeding response	16 w	No	[65]
MYH9-RD (1)	Case report	Romiplostim	6.3 mcg/kg/w	6 × 10^9^/L	Misdiagnosis of ITP (3th lines)	PR	41 m	No	[64]
DiGeorge syndrome (1)	Case report	Eltrombopag	25–50 mg/d	2 × 10^9^/L	Misdiagnosis of ITP (4th lines)	CR	13 m	No	[66]
PT syndrome (1)	Case report	Eltrombopag	50–150 mg/d	9–20 × 10^9^/L	Misdiagnosis of ITP (1st line)	PR and no bleeding	23 m	No	[67]
WAS (1)	Case report	Eltrombopag	0.8–5 mg/kg/d	10 × 10^9^/L	Bridging to HSCT	PR and reduced bleeding and platelet transfusion	32 w	No	[68]
WAS (1)	Case report	Eltrombopag	25 mg/d	20 × 10^9^/L	Engraftment failure post HSCT	NR	7 m	No	[66]
THPO-RT (2)	Case series	Romiplostim	5 mcg/kg/w	Both 21 × 10^9^/L	BMF & HSCT failure	CR and no bleeding	2 y 13 w	No	[69]
THPO-RT (3)	Case series	Romiplostim	4 mcg/kg/m	3–27 × 10^9^/L	BMF & Transfusions	CR and no bleeding	>6 y	No	[70]

Abbreviations: PT: Paris-Trousseau; RD: related disorder; WAS: Wiskott–Aldrich Syndrome; n: number of patients included; y: years; F: female; M: male; PC: platelet counts; bleeding was defined according to WHO bleeding scale; y: years; m: month; w: weeks; d: day; Ref: reference; RT: related thrombocytopenia; BMF: bone marrow failure; HSCT: hematopoietic stem cell transplant; ITP (lines): n° of lines used previous to TPO-RA; CR: complete response; PR: partial response; NR: no response.

## Data Availability

Not applicable.

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
