# Peer review of "Role of Thrombopoietin Receptor Agonists in Inherited Thrombocytopenia"

_ijms, 2021, doi:10.3390/ijms22094330_

Round 1

Reviewer 1 Report

the work is well structured and interesting from a scientific point of view. The various sections are clear and the literature search good.

Author Response

Thank you very much for your comments

Reviewer 2 Report

This is very interesting review paper which demonstrate molecular aspects of the role of thrombopoietin receptor agonists in inherited thrombocytopenia. The paper has been very carefully thought and the conception demonstrated experience of Authors in such topic. I have only suggestion for tables 2-4 in my opinion should be corrected to me more legible.

Author Response

Thank you very much for your comments. We have deleted some information in the tables to me more legible.

Reviewer 3 Report

Authors present a well written and interesting review article. The use of TPO-RA in inherited thrombocytopenias could offer an important alternative treatment both for bleeding episodes and as prophylaxis before operations. At this time, the use is off-label and a review of reported cases is useful in clinical practice.

Author Response

Thank you very much for your comments